# The Quality of Telenursing—Israeli Nursing Staff’s Perceptions

**DOI:** 10.3390/healthcare11222915

**Published:** 2023-11-07

**Authors:** Keren Grinberg, Yael Sela

**Affiliations:** Department of Nursing Sciences, Faculty of Social and Community Sciences, Ruppin Academic Center, Emek-Hefer 4025000, Israel; yaels@ruppin.ac.il

**Keywords:** telemedicine, telenursing, nurses, quality of care

## Abstract

Background: The outbreak of the COVID-19 pandemic has increased telemedicine and telenursing services worldwide, developed this innovative treatment’s potential, and emphasized its importance. The constraints imposed by the pandemic breached regulatory, psychological, and organizational obstructions among both patients and caretakers. Community and hospital nursing services were forced to deal with a new reality, to provide remote care solutions for bedridden chronic patients, as the need for this grew exponentially. Despite the increase of telemedicine in recent years, so far no research in Israel has investigated the nursing staff’s perceptions of the quality of the care provided through telenursing. Objective: To assess nurses’ perceptions of the quality of the care provided through telenursing compared to face-to-face nursing. Method: A quantitative cross-sectional study among 227 male and female nurses in Israel. The questionnaire included demographic questions, and the five measures of quality of care: concern and empathy for the patient, professional treatment, response to treatment, educated use of resources, and patients’ sense of security. Results: Significant differences were found between nurses’ perceptions of telenursing and face-to-face nursing. The quality of face-to-face nursing was perceived as more positive than that of telenursing, in general, as were the individual measures of care quality such as providing professional treatment, response to treatment, and patients’ sense of security. Conclusions: Although telemedicine has increased significantly in recent years, nursing staff still perceived the quality of care and follow-up of face-to-face nursing as more positive. It is important to continue to monitor nurses’ perceptions and attitudes towards the strengths of these two treatment methods, not only in crisis situations, but among wider populations, and to investigate the factors that could influence these perceptions.

## 1. Introduction

Telemedicine refers to all encounters between patients and medical staff in which the parties are not in the same location [1]. The connection is accomplished using technological means such as telephone, text message, email, chat, and video call. Telenursing is a subgroup of telemedicine and is defined as providing remote nursing services that include treatment management, guidance, and coordination for the patient and family through technological and digital means as an alternative to a face-to-face session [2,3]. 

The norms that bind nurses in any treatment framework, i.e., their authority by virtue of Ministry of Health licensing, CEO circulars, internal regulations of hospitals and HMOs, etc., do not differentiate between face-to-face and distant nursing. Nurses are expected to provide quality care, maintain a high professional standard, and pay comprehensive attention to the patient and his/her family [4]. Additionally, the Israeli Ministry of Health’s CEO circular of June 2019 that determined the standards for operating telemedicine noted that on whichever channel care is provided, service providers should recognize their professional and legal responsibility for the actions of telemedicine, and undergo appropriate training [5]. 

The possibilities inherent in telemedicine create advantages on three main levels. First, it provides accessibility to treatment for patients who have so far avoided using face-to-face health services. According to the American Telemedicine Association, telenursing is a good solution primarily for the housebound and elderly population, providing them with maximum security and comfort, and even increasing their cooperation and responsibility for their care [6,7]. Telenursing has also been found to have great advantages for housebound Chronic Obstructive Pulmonary Disease (COPD) patients [8,9]. Second, the use of technology and distant services empowers nurses and helps develop their role. Nurses can oversee, educate, follow up, and provide multidisciplinary treatments, including pain management and support for patients and families [10]. Third, health systems benefit by reducing the number of ‘no-shows’, improving the nurse-patient ratio, and radically cutting costs [1,6]. Moreover, telemedicine helps the health system face the complex challenges of aging populations, increased chronic illness, and lack of manpower. In view of the need to deal with high healthcare costs, on the one hand, and increased demand for provision and accessibility of services, on the other hand, it is expected that the use of telemedicine will intensify [7,8,11]. 

However, alongside the advantages of telemedicine, it encompasses quite a few medical, social, and economic difficulties and challenges. First, on the policy level, it is feared that HMOs may inefficiently allocate many resources for the new telemedicine technologies in an attempt to attract a younger, healthier, and more profitable population [1]. Also, difficulties in accessing technological devices, as well as lack of knowledge, control, and operation of technological equipment, could block treatment for both patients and their families, and nurses [12,13]. Reiss et al. [14], who examined the feasibility of using distant technological means during the COVID-19 pandemic, found that telemedicine is not suitable for everyone, with specific difficulties found among the over-65 adult population. 

Furthermore, nurses reported that using technology could undermine their confidence in the question of whether their professional qualifications and skills, which were acquired through face-to-face care, would suffice to cope with the unfamiliar ground in a way that allows them to provide high-quality professional care, and to implement their and their colleagues’ authority. The nurses’ role and training rely greatly on ‘soft-core skills’ such as empathy, communication, and trust. These are skills that nurses employ in frontal meetings, but their implementation in telenursing is complex and necessitates specific relevant training [15]. For instance, identifying signs of abuse, which requires attention to body language, often hinges on the nurse’s intuition, and depends on the personal relationship between the nurse and patient. There is concern that in remote services these core skills would be inoperable, and optimal care would be impossible. Another significant difficulty in telemedicine is building trust with the patient, which is necessary for the treatment’s success. Also, nurses are required to have high social and interpersonal skills, and communication skills to provide quality care similar to their face-to-face sessions [15]. Ethical and moral issues of telenursing also raise concerns among nurses. For example, receiving the patient’s informed consent when surrounded by family members, recording or misuse of the session content, and aspects of privacy. The limitations presented by telenursing could damage the nurses’ ability to provide professional and high-quality care [16]. 

The option of delivering remote care through various technological means has been in use in different countries for a few years. Telenursing, or remote nursing care, has proven to be a valuable approach in modern healthcare. However, there are alternative methods and approaches to providing nursing care when telenursing is not feasible or appropriate. Some of the alternatives to telenursing include face-to-face nursing care, home health care, telehealth with video conferencing, telephone consultations, mobile health apps’, etc. [2,7,12]. The COVID-19 pandemic outbreak has dramatically intensified the use of technologies (such as video calls) in many countries including Israel. The constraints created by the pandemic have breached regulatory, psychological, and organizational barriers, among both patients and caregivers, and the use of telemedicine has increased considerably [17]. An estimate in Israeli HMOs indicated that in March-April 2020, clinic visits decreased by 50% compared to January-February 2020. It should be noted that some of the visits were replaced by telemedicine, mainly telephone appointments or email correspondence [13]. A study among 169 pediatricians in Israel and analysis of Maccabi HMO data found that within a few days, the use of telephone appointments skyrocketed from zero to 2000 appointments per day and that the use of telephone appointments did not decline during the post-lockdown period, but increased to 3000 appointments per day [1]. 

With the COVID-19 outbreak, community nursing services in all HMOs and various hospitalization frameworks were expected to provide solutions for home-ridden chronic patients, which increased the use of video calls with chronic patients such as blood pressure monitoring or insulin injection instructions, oncological follow-up, medication assessment and guidance in a variety of cases, diabetic wounds follow-up, and pregnancy management and accompaniment [18,19]. 

The health system’s ability to integrate the use of technology among nurses depends greatly on the nurses’ perception of the quality of the care they provide on all channels. ‘Quality of care’ is described as the degree to which the patient’s physiological, psychological, and social needs are met by the nurses, and the degree of the therapeutic effect of their care on the patient’s recovery [10,20]. 

To succeed in providing remote quality care that meets the patient’s needs, and to express their professional knowledge, interpersonal empathy, and communication skills, which they were trained for, nurses must be understanding of the needed technical requirements and ‘soft skills’, see the advantages of using them for both themselves and their patients, and feel confident that they can realize their authority and provide quality care that meets the comprehensive needs of the patients [21]. In view of the fact that, on the one hand, some of the most important interpersonal skills are not evident in telenursing and, on the other hand, nurses are committed to a uniform professional standard on the various treatment channels, it is suitable to examine nurses’ perceptions of quality of care and to compare them. When nurses feel that their work conditions allow them to be safe and protected by the law, in a way that allows them to provide optimal quality care, and that they can preserve and even improve their skills in telenursing, the assimilation of these uses among nurses will increase, and telenursing will become an inseparable part of their toolbox. 

The CEO’s circular on telemedicine [5] mentions that the meeting presented by telemedicine requires extra caution when determining the diagnosis and treatment. It continues to say that telemedicine requires unique training for all administrators and caregivers in the service. The main form of training that is recommended relates to aspects of professionalism, use of technology, and legal issues. But, on the topic of nurse-patient relationships, the report determined that the training should include: an explanation of how to check that the medical instructions were understood by the patient, aspects of medical confidentiality, and does not refer to the training nurses need to exercise their authority from a distance, both safely and qualitatively. Since the scope of telenursing has also grown considerably, we see a need for an empiric study—the first of its kind in Israel—to examine nurses’ perceptions of the difficulties and challenges and shed light on their perceptions of the quality of care provided by nurses [13]. 

Nursing is a crucial and significant part of the health services in Israel and, as such, must play a significant role in providing solutions for the challenges faced by the health system [22]. The 21st-century nursing paradigm requires changing the treatment strategy concerning the efforts to continue to provide patients with high-quality bio-psycho-social care, and digital tools are becoming an integral part of it. It is, therefore, important to understand the nurses’ perception of the quality of care in depth. Functional policy to integrate the use of telenursing should be planned, based on careful analysis and understanding of the barriers. After reviewing previous studies, it became evident that there was a lack of research regarding the differences in perceptions between face-to-face nursing and remote nursing. Furthermore, the emergence of the COVID-19 crisis has underscored the need for a fresh perspective on this matter.

Therefore, the aim of this study was to assess the perception of the quality of telenursing in comparison to face-to-face nursing among nurses. The following hypotheses were formulated:

**H1.** 
*Differences will be found between nurses’ perceptions of the quality of care of face-to-face nursing and telenursing so that their assessment of face-to-face nursing will be more positive than their assessment of telenursing.*


**H2.** 
*The five measures of the quality of nursing care, concern and empathy for the patient, providing professional treatment, response to treatment, educated use of resources, and patients’ sense of security, will be more positive in face-to-face nursing than in telenursing.*


## 2. Method

This is a quantitative cross-sectional study.

### 2.1. Tools

An online questionnaire was built, based on a well-known questionnaire that examined perspectives of telemedicine efficacy and quality of health care among patients and professionals. It was constructed based on the literature review [10,23] and was first administered as a pilot among 10 nurses. Corrections were made as per their recommendations. Additionally, content validity was supported by a panel of experts who reviewed the questionnaire twice in order to determine the representativeness and relevance of the items. The questionnaire included measures of physical aspects of service, empathy, security, trustworthiness, and responsiveness, and helped to thoroughly assess the nurses’ viewpoint. The questionnaire was expanded based on the relevant literature to examine perceptions of remote nursing quality, in a way that combines the measures that examine nurses’ perceptions of quality with their feelings about implementing it through telenursing, and how these can be integrated. For example, aspects of their ability to fulfill patients’ needs or implement ‘soft skills’ in telenursing. 

The first part of the questionnaire included 13 demographic questions (such as age, gender, education, place of employment, whether s/he had worked in a COVID unit, etc.). The second part included 18 statements on a scale of 1 (completely disagree) to 5 (completely agree) relating to the quality of the care provided to patients (e.g., “Telenursing is more accessible for patients than face-to-face nursing”). This section was divided into five measures: concern and empathy for the patient, professional treatment, treatment compliance, educated use of resources, and patients’ sense of security. The general reliability of the entire questionnaire was Cronbach’s α = 0.872 and its validity was received by content validity. 

### 2.2. Participants

A total of 227 nurses employed in various medical centers answered the questionnaire. The sample included 59 men and 168 women aged 18–60 who work in the health system, in hospitalization institutions, and in public health centers (community health centers). The respondents were located through a convenience sample, which is based on choosing the easiest respondents to approach.

### 2.3. Procedure 

The study was approved by the ethics committee of the Ruppin Academic Center. Following approval, the online questionnaire was distributed on social networks and nursing forums through snowball sampling during January–March 2022 (parallel to the fifth wave of COVID). The questionnaire was anonymous, and the participants were assured that their privacy would be maintained throughout the study. The questionnaire was distributed with a link and an explanation of the study and its purpose. An informed consent form was signed by all participants. Completion of each questionnaire took approximately 15 min. 

### 2.4. Data Analysis

The data were processed and analyzed with SPSS (version 26) statistical software. Data were analyzed by using descriptive statistics followed by the appropriate inferential statistics. T-tests were employed to examine the differences between nurses’ perceptions of quality of care of face-to-face nursing and telenursing and also for the differences between nurses’ perceptions of the five measures of quality of care of face-to-face nursing and telenursing. 

## 3. Results 

The research population included 227 nurses, most of whom (64.3%) worked in hospitals. Slightly over half of the respondents (54.6%) had experience in telenursing, and the majority (73.1%) had a Bachelor’s degree. Most of the respondents (91.2%) mentioned that during the COVID-19 pandemic the use of telenursing had increased considerably. The respondents’ characteristics are featured in Table 1. 

As mentioned, the quality-of-care index includes five measures: concern and empathy for the patient, professional treatment, response to treatment, educated use of resources, and patients’ sense of security. The highest measure was the educated use of resources (M = 3.319). The results of the respondents’ assessments are depicted in Table 2.

To examine the first hypothesis, that differences would be found between nurses’ perceptions of quality of care of face-to-face nursing and telenursing, a t-test for independent samples was conducted. The dependent variable was the quality of care, and the independent variables were face-to-face nursing and telenursing. The results are shown in Table 3. 

The results showed a significant difference in the perception of quality between telenursing and face-to-face nursing (*t* [227] = 4.570, *p* < 0.01). That is to say, nurses’ perception of the quality of care given to patients was higher for face-to-face nursing than for telenursing. Hypothesis H1 was corroborated.

To examine the second hypothesis, that the five measures of quality of nursing care would be more positive in face-to-face nursing than in telenursing, a t-test for independent samples was performed. Table 4 shows the results.

Table 4 indicates that H2 was partially corroborated. The results showed that the nurses’ perceptions of four (out of five) measures were significantly more positive for face-to-face nursing than for telenursing. The difference between telenursing and face-to-face nursing for the measure of ‘educated use of resources’ was non-significant.

## 4. Discussion

The main goal of this study was to examine nursing staff’s perception of their quality of care, and whether there would be a difference between face-to-face nursing and telenursing. In general, the results indicated that there were significant differences in the perceived quality of care and its measures between face-to-face nursing and telenursing.

Nurses, as change agents, have a share in various fields, including the use of appropriate technology for the nursing care process. Nevertheless, it has been limited due to the physical condition, burden, location of the patient, and geographic location, while the current instructions for the implementation and maintenance of health must be patient-centered [24]. The use of technology is a very appropriate strategy. It is needed to meet the needs of care in a sustainable manner and provide a new window for improving care services [25]. However, the existence of telenursing can improve health services by creating new and innovative models of care which must consider the fatigue factor of nurses, the needs of the millennial workforce, and how to balance the skills of new graduates and experienced care in the hospital. Studies revealed that while the demand for telehealth care and involvement of nursing staff is increasing, knowledge of factors that influence nurses’ intention and willingness to practice telenursing is limited [26,27].

H1 was corroborated, supporting the hypothesis that nurses’ assessment of the quality of care of face-to-face nursing would be more positive than their assessment of telenursing. It may be that the elements of knowledge and skill affected the nurses’ perceptions of the care and its quality.

The second hypothesis was not fully substantiated. It was found that the measures of quality of care, providing professional treatment, responsiveness to care, and patients’ sense of security were higher for face-to-face nursing than for telenursing. The literature has shown that difficulty in accessing technology, and a lack of knowledge to control and operate technological systems, could be a barrier to telenursing for both the nurse and the patient and his/her family [12]. In addition, patients’ difficulties in activating technological tools could prevent them from following the nurse’s instructions or recommendations, which could, in turn, deteriorate their medical condition and make telenursing ineffective [28,29]. That is to say, the use and operation of technology to provide distant care requires technological understanding and skills, which could affect the nurses’ perception of telenursing. The present findings support Gidora et al. [30], who reported that clinical decision support systems reflect the fact that nurses are unable to see their clients as well as not being able to collect objective health data and may recommend higher levels of care than a clinician would recommend in a face-to-face encounter.

An interesting finding concerns the measure ‘concern and empathy for patients’, for which we found a significant difference between face-to-face and telenursing, but in the opposite direction; namely, concern and empathy were higher for telenursing than for face-to-face nursing. It may be that treating from a distance could make nurses think that there were medical issues that had not been identified or treated. There could also be the fear of misdiagnosis, which could lead to more concern and empathy. Furthermore, the need to demonstrate empathy could be stronger in telenursing when the nurse is not physically near the patient and cannot touch him/her. This finding is in line with Hogan et al. [21], who clearly indicated that comprehensive high-quality telenursing service requires empathy, concern, and compassion for the patient, certainly no less than in face-to-face treatment.

Telenursing triage and advice services have the potential to save money and resources within the overall healthcare system [30]. However, as opposed to the advantages mentioned in the literature [6,31], the present study found no significant difference in the nurses’ perceptions of the measure ‘educated use of resources’.

As mentioned, a majority of respondents (91.2%) said that during the COVID-19 pandemic, the use of telenursing had increased considerably. This corresponds with the State Comptroller’s report [13] which argued that the outbreak of the pandemic had dramatically increased the use of technologies because of the constraints created by the pandemic. An assessment by Israeli HMOs revealed that in March–April 2020, the number of physical visits to clinics had fallen by 50% compared to January–February of the same year (before the pandemic). Some of these visits had been converted to telemedicine, mainly by phone or e-mail. Both of these sources confirm the respondents’ perceptions.

Due to COVID-19 restrictions, face-to-face service was unfeasible, especially for at-risk groups, which created the need to employ telenursing to maintain treatment continuity. Notwithstanding, telemedicine is not suitable for every medical condition [32]. Also, it is impossible to remotely perform examinations that require physically touching the patient’s flesh or skin [33]. However, telenursing allows nurses to communicate with patients remotely, maintain continuity of treatment, provide guidance, and make sure that they get the necessary care. It can be concluded that telenursing cannot be a total replacement for face-to-face treatment but can support and reinforce its strengths. As we experienced during the COVID-19 pandemic, telenursing has the potential for a variety of situations of inaccessibility and geographic distance, can provide solutions for at-risk populations at times of crisis, and, as such, should be included in plans for future pandemics.

The nurse, as a subject, has a comprehensive role, including the role of developing theories and models in nursing. The philosophical and contextual issues in nursing theory in the 21st century have also been developed in the theory of caring models in technology-based care. Nurses, in carrying out nursing care, will have a lot of progress, innovation, and effectiveness in responding to social demands for a paradigm shift in nursing care that leads to the need for satisfaction for family and community patients [34]. The development of technology-based treatment methods and telemedicine have played an important role in coping with the COVID-19 pandemic. However, this was a worldwide crisis, that required swift adjustments and immediate solutions, so nurses did not use it by preference or choice but by necessity [35].

## 5. Conclusions

Telenursing have gained widespread acceptance in various healthcare systems, becoming standard tools for healthcare providers. The advantages of telenursing were further underscored during the COVID-19 pandemic, particularly in light of the ‘social distancing’ guidelines, which made in-person patient interactions challenging. By leveraging technology, nurses can oversee, educate, follow up, and provide comprehensive care, encompassing pain management and support for both the patient and their family [10], all while mitigating the risk of contagion. Nevertheless, it appears that nurses generally hold a more positive perception of the quality of care in face-to-face nursing. Current findings suggested that nurses view in-person nursing as more beneficial for patients and of higher quality, covering three dimensions of care quality: professional treatment, responsiveness to treatment, and patients’ sense of security.

This study represents a pioneering exploration, and to the best of our knowledge, no prior research has delved into the five aspects of care quality and how nurses perceive them when comparing the two treatment methods. It remains crucial to continuously scrutinize nursing staff’s perceptions and approaches concerning the strengths of both methods and the five dimensions of care quality, not solely during crises, but also as a routine practice among a broader demographic. Equally important is the examination of factors influencing nurses’ perceptions regarding telenursing and face-to-face nursing.

In light of the complexity of the pandemic and the necessity to adapt to changes in the healthcare system, nurses found themselves navigating the fine line between fulfilling their professional and ethical responsibilities by providing remote care. Simultaneously, their limited experience and knowledge in telenursing, including specialized training, may have contributed to ambivalence in their perceptions. It is imperative to continue exploring nurses’ views on telenursing, especially in cases where they receive specific training for remote treatment.

## 6. Limitations

This study had a few limitations. First, the study was not probabilistic. The questionnaire was distributed on internet forums and social media, and the sampling method was the snowball method. As such, the sample does not necessarily represent the general population of nurses, and the findings cannot be generalized to all nurses in Israel. Second, the findings may not necessarily reflect the reality. The data was based on nurses’ self-reports and subject to ‘social-desirability bias’, namely, people’s tendency to present themselves in a generally favorable fashion.

Furthermore, not all of the respondents had experienced telenursing, so their answers could have been based on general knowledge and/or their colleagues’ opinions. Also, the timing of the questionnaires, quite close to the COVID-19 pandemic, could have affected the nurses’ answers. Therefore, the issue should be examined again after some time, once the system has returned to routine, to see whether the nurses’ perceptions have changed.

## Figures and Tables

**Table 1 healthcare-11-02915-t001:** Sociodemographic characteristics of the sample (*n* = 227).

Variable		*n*	%
Gender	Male	59	26.0%
	Female	168	74.0%
Age	20–40	132	58.1%
	41–50	75	33.0%
	51–60	20	8.8%
Education	High school/tertiary	14	6.2%
	First degree	166	73.1%
	Second degree	39	17.2%
	Third degree	8	3.5%
Place of employment	Hospital	146	64.3%
	Private health center	22	9.7%
	Community	59	26.0%
Nursing experience (in years)	0–2	66	29.1%
	3–5	29	12.8%
	6–10	35	15.4%
	Over 10	97	42.7%
Telenursing experience	Yes	124	54.6%
	No	103	45.4%
Work/ed in COVID unit	Yes	86	37.9%
	No	141	62.1%
Field of telenursing care	Pregnancy	13	5.7%
	Diabetes	36	15.9%
	Wound	19	8.4%
	COVID	83	36.6%
	Other	76	33.5%
Telenursing experience (in years)	0–2	177	78.0%
	3–5	23	10.1%
	6–10	10	4.4%
	Over 10	17	7.5%

**Table 2 healthcare-11-02915-t002:** Means and ranges of quality of nursing measures.

Variable	Mean (SD)	Range
General	2.930 (0.633)	1.44–4.78
Concern and empathy for patients	3.018 (0.814)	1.00–5.00
Providing professional treatment	2.728 (0.634)	1.00–4.50
Responsiveness to treatment	3.032 (0.754)	1.00–5.00
Educated use of resources	3.319 (0.786)	1.00–5.00
Patients’ sense of security	2.771 (0.853)	

**Table 3 healthcare-11-02915-t003:** Differences between nurses’ perceptions of quality of care of face-to-face nursing and telenursing.

	Telenursing (*n* = 100)	Face-to-Face Nursing (*n* = 127)	*t* [227]
Mean	2.766	3.137	4.570 **
(SD)	(0.614)	(0.601)	

** *p* < 0.01.

**Table 4 healthcare-11-02915-t004:** Differences between nurses’ perceptions of the five measures of quality of care of face-to-face nursing and telenursing.

	Telenursing (*n* = 100)	Face-to-Face Nursing (*n* = 127)	*t* [227]
Concern and empathy for the patient	3.292	2.803	4.969 **
	(0.735)	(0.811)	
Professional treatment	2.579	2.917	4.128 **
	(0.622)	(0.599)	
Response to treatment	2.879	3.227	3.534 **
	(0.690)	(0.790)	
Educated use of resources	3.430	3.232	1.892
	(0.810)	(0.758)	
Patients’ sense of security	2.602	2.985	3.433 **
	(0.829)	(0.839)	

** *p* < 0.01.

## Data Availability

The data presented in this study are available on request from the corresponding author.

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
