# Peer review of "The Quality of Telenursing—Israeli Nursing Staff’s Perceptions"

_healthcare, 2023, doi:10.3390/healthcare11222915_

Round 1

Reviewer 1 Report

Comments and Suggestions for Authors

Overall, the researcher's logical writing was excellent.

I also believe that telenursing is a direction in nursing that will definitely be implemented in future society. Therefore, I am confident that it is a good research topic.

1. It is recommended that references from 2005 and 2008 be deleted or changed to the latest version.  

2. The research hypothesis presented by the researcher is that face-to-face nursing is generally recognized as better than remote nursing. It is necessary to clearly state in the introduction the basis for why nurses' perceptions of this should be investigated. For example, 1) as a result of a review of previous research, no research has been conducted on the differences in perception of face-to-face nursing and remote nursing, or 2) no consistent results have been reported as a result of prior research, or 3) a situation in which a new perspective is required.  

3. The research methodology is excellent. However, in order for this study to be established, the basis for establishing the hypothesis must be sufficiently presented in the introduction.  

4. Finally, in the discussion, a comparative analysis needs to be made between the results of this study and various (rich) previous studies. After comparative analysis with previous research results, it would be good to clearly describe the significance of this research result. Currently, I think this part is described somewhat unclearly.

Author Response

The authors thank the editor and the reviewers for their important comments and their review. We have addressed each of the new comments and corrected the manuscript accordingly. The corrections were made by the "Track Changes" function in Microsoft Word, so that changes are easily visible to the editor and reviewers. Our response is highlighted in green to make it easier to read.

Reviewer 1

Overall, the researcher's logical writing was excellent.

I also believe that telenursing is a direction in nursing that will definitely be implemented in future society. Therefore, I am confident that it is a good research topic.

  1. It is recommended that references from 2005 and 2008 be deleted or changed to the latest version.  

Response: Thank you. The source from 2005 is one of the only ones that made this claim, so it was not deleted, and the source from 2008 is related to the research tool, so we kept it. 

  1. The research hypothesis presented by the researcher is that face-to-face nursing is generally recognized as better than remote nursing. It is necessary to clearly state in the introduction the basis for why nurses' perceptions of this should be investigated. For example, 1) as a result of a review of previous research, no research has been conducted on the differences in perception of face-to-face nursing and remote nursing, or 2) no consistent results have been reported as a result of prior research, or 3) a situation in which a new perspective is required.  

Response: Thank you for this important comment. We have added this relevant information in the introduction section (line 152-153, page 3-4).

  1. The research methodology is excellent. However, in order for this study to be established, the basis for establishing the hypothesis must be sufficiently presented in the introduction.  

this study and various (rich) previous studies. After comparative analysis with previous research results, it would be good to clearly describe the significance of this research result. Currently, I think this part is described somewhat unclearly.

Response: Thank you So far, to the best of our knowledge, not many studies have been done on the subject, but articles discussing this comparison have been included in the introduction and discussion sections.

Reviewer 2 Report

Comments and Suggestions for Authors

The theme of the article is important to reflect on, especially after being confronted with the pandemic. However, the article raised doubts that may be important to clarify in order to make the study more robust:

What are the reasons that led to teleconsultations? Was there an alternative?

The methodology describes that the sample consisted of 227 nurses with experience in telecare (inclusion criterion). The results show that this criterion was not met, so the bias appears from the outset. The authors should remove this inclusion criterion (line 159).

Have the nurses with experience been trained?

Can the quality of care be "measured" by the response to treatment? Wouldn't it be wiser to replace this item with "adherence to the therapeutic regimen" (line 176-178...)?

Author Response

The authors thank the editor and the reviewers for their important comments and their review. We have addressed each of the new comments and corrected the manuscript accordingly. The corrections were made by the "Track Changes" function in Microsoft Word, so that changes are easily visible to the editor and reviewers. Our response is highlighted in green to make it easier to read.

The theme of the article is important to reflect on, especially after being confronted with the pandemic. However, the article raised doubts that may be important to clarify in order to make the study more robust:

What are the reasons that led to teleconsultations? Was there an alternative?

Response: The authors have added this relevant information in the introduction section. (line, 91-96 page 2).

The methodology describes that the sample consisted of 227 nurses with experience in telecare (inclusion criterion). The results show that this criterion was not met, so the bias appears from the outset. The authors should remove this inclusion criterion (line 159).

Response: Thank you for this comment. The authors deleted this to avoid bias (line 168-170, page 4).

Have the nurses with experience been trained?

Response: This is a very important question. Unfortunately, the nurses were not asked this question. It is important to check this in follow-up studies. We have added it to the recommendations (line 353-359, page 8).

Can the quality of care be "measured" by the response to treatment? Wouldn't it be wiser to replace this item with "adherence to the therapeutic regimen" (line 176-178...)?

Response: Thank you for this important comment. Indeed, the intention is to adhere to the treatment, the intention is to compliance. The wording has been changed in order to clarify this (line 190, page 4).

Reviewer 3 Report

Comments and Suggestions for Authors

The article's subject is quite interesting, but chapter “I. Method” needs to be completely rewritten because it's not exhaustive; even the design of the study is missing, an absolutely essential element to indicate in a paper. The tool is the most vulnerable element of this study.

Tools: The instrument's explanation is incredibly lacking and needs to be expanded upon. Detail the tool's questions in a table or an appendix and explain according to which evidence-based indications it was developed. Additionally, because this is an ad hoc, self-created questionnaire, indicate that it's not a validated tool and that you are unsure of its validity and feasibility; as a consequence, the results may be seriously skewed because no one can be certain that the instrument accurately measures the phenomenon in question (this can only be known after having conducted a validation study of the questionnaire; if it has been carried out, indicate the results).

Best regards.

Author Response

The authors thank the editor and the reviewers for their important comments and their review. We have addressed each of the new comments and corrected the manuscript accordingly. The corrections were made by the "Track Changes" function in Microsoft Word, so that changes are easily visible to the editor and reviewers. Our response is highlighted in green to make it easier to read.

Response: Thank you! We edited the methods section and added relevant information regarding the design and the tool and data collection and analysis (page 4).

Reviewer 4 Report

Comments and Suggestions for Authors

Very interesting topic at the moment, well justified and well supported by scientific evidence in the introduction.

The methodology does not present the type of study and sample. It does not describe the questionnaire rigorously and systematically. They don't explain most of the methodological options (data collection) and they don't clearly justify the ethical issues involved in a study that collects data on social networks and forums. They were poor at processing the data and wasted a lot of sociodemographic variables. This implies a major review of methodological issues and results.

The results, with a good comparison of authors, were in line with the hypotheses and general objective of the study, but could be further developed.

The conclusions should clearly respond to the aim of the work. The first part (first paragraph) can be more summarized and does not need to have references to authors.

A great detail and clarity in the limitations, it's a pity they didn't prevent some aspects during the study. Some references with details to be corrected by Vancouver. There are authors supporting Telenursing and telemedicine that need to be updated (2009, 11, 12... etc).

Author Response

The authors thank the editor and the reviewers for their important comments and their review. We have addressed each of the new comments and corrected the manuscript accordingly. The corrections were made by the "Track Changes" function in Microsoft Word, so that changes are easily visible to the editor and reviewers. Our response is highlighted in green to make it easier to read.

Very interesting topic at the moment, well justified and well supported by scientific evidence in the introduction.

The methodology does not present the type of study and sample. It does not describe the questionnaire rigorously and systematically. They don't explain most of the methodological options (data collection) and they don't clearly justify the ethical issues involved in a study that collects data on social networks and forums. They were poor at processing the data and wasted a lot of sociodemographic variables. This implies a major review of methodological issues and results.

Response:  Thank you. We edited the methods section and added relevant information regarding the design and the tool (page 4-5).

The results, with a good comparison of authors, were in line with the hypotheses and general objective of the study, but could be further developed.

Response: Thank you. If it is necessary to present additional findings, the authors will be happy to know what exactly is required.

The conclusions should clearly respond to the aim of the work. The first part (first paragraph) can be more summarized and does not need to have references to authors.

Response: Thanks for this comment. The conclusion paragraph has been rewritten (page 8).

A great detail and clarity in the limitations, it's a pity they didn't prevent some aspects during the study. 

Response: That is correct and there are recommendations for further research.

Some references with details to be corrected by Vancouver. There are authors supporting Telenursing and telemedicine that need to be updated (2009, 11, 12... etc).

Response:  Thank you. That was corrected.

Round 2

Reviewer 1 Report

Comments and Suggestions for Authors

Thank you for your efforts to make changes to reflect the reviewer's opinions.

Reviewer 3 Report

Comments and Suggestions for Authors

The authors' additions are appropriate.

Best regards.

Reviewer 4 Report

Comments and Suggestions for Authors

As I mentioned earlier, this is a very interesting topic for guaranteeing healthcare that is closer and more accessible to citizens and for increasing the quality of current healthcare practices.

Most comments and suggestions were responded to positively.

The scientific methodology has become clearer, with a more detailed and systematic description of the procedures carried out.

I would like to congratulate the authors on their prompt and effective response to the article, having valued its quality, both for the description given and for the improved clarity of the procedures and ethical principles in carrying out the data collection.

Throughout the text, especially in the conclusion, the words telemedicine and telenursing still appear indistinctly, the terms could be revised and uniformised.